# The Role of T-Cadherin (CDH13) in Treatment Options with Garcinol in Melanoma

**DOI:** 10.3390/cancers16101853

**Published:** 2024-05-12

**Authors:** Sebastian Staebler, Sebastian Hoechst, Aranya Thongmao, Nadja Schneider, Anja-Katrin Bosserhoff, Silke Kuphal

**Affiliations:** Institute of Biochemistry, Friedrich Alexander University Erlangen-Nürnberg, Fahrstrasse 17, 91054 Erlangen, Germany; sebastian.staebler@fau.de (S.S.); sebastian.hoechst@fau.de (S.H.); aranya.thongmao@fau.de (A.T.); nadja.schneider@fau.de (N.S.); anja.bosserhoff@fau.de (A.-K.B.)

**Keywords:** CDH13, T-cadherin, garcinol, malignant melanoma

## Abstract

**Simple Summary:**

Malignant melanoma is the deadliest form of skin cancer. Since resistance and relapse to established therapy regimes for the treatment of malignant melanoma still occur frequently, the development of novel therapeutics is the key to improve patient prognosis. The study aimed to understand the function of the natural compound Garcinol on melanoma cells to contribute to its potential use in tumor therapy. We could show that Garcinol reduced proliferation and induced apoptosis in melanoma cells. Furthermore, we showed that melanoma cells expressing T-cadherin (CDH13) respond more sensitively to garcinol treatment. We postulate, that particularly T-cadherin-positive-patients could be receptive to garcinol therapy.

**Abstract:**

Targeted therapies with chemotherapeutic agents and immunotherapy with checkpoint inhibitors are among the systemic therapies recommended in the guidelines for clinicians to treat melanoma. Although there have been constant improvements in the treatment of melanoma, resistance to the established therapies continues to occur. Therefore, the purpose of this study was to explore the function of garcinol with regards to specific cancer properties such as proliferation and apoptosis. Garcinol, a natural compound isolated from the plant also known as mangosteen (*Garcinia mangostana*), is a newly discovered option for cancer treatment. Numerous pharmaceutical substances are derived from plants. For example, the derivates of camptothecin, extracted from the bark of the Chinese tree of happiness (*Camptotheca acuminate*), or paclitaxel, extracted from the bark of the Western yew tree (*Taxus brevifolia*), are used as anti-cancer drugs. Here, we show that garcinol reduced proliferation and induced apoptosis in melanoma cell lines. In addition, we found that those cells that are positive for the expression of the cell–cell adhesion molecule T-cadherin (CDH13) respond more sensitively to treatment with garcinol. After knock-down experiments with an siRNA pool against T-cadherin, the sensitivity to garcinol decreased and proliferation and anti-apoptotic behavior of the cells was restored. We conclude that patients who are T-cadherin-positive could especially benefit from a therapy with garcinol.

## 1. Introduction

Melanoma is a deadly form of skin cancer with high rates of resistance to traditional chemotherapy and radiotherapy. BRAF inhibitors (BRAFis) and MAPKinase inhibitors (MAPKis) can achieve initial efficacy when used to treat melanoma patients. However, drug resistance and relapse are common, emphasizing the need for new therapeutic strategies [1]. Existing BRAFi/MAPKi therapies lose effectiveness through activation of substitutive pathways like PI3K/AKT, NF-κB or EGFR [2]. Therefore, entirely new compounds also need to be investigated. Garcinol, a new chemically synthesized polyisoprenylated benzophenone, is the medicinal compound obtained from fruits and leaves of the plant *Garcinia indica*, which was first extracted in 1981 [3]. A review published in 2021 summarizes the potency of garcinol to inhibit histone acetyltransferases (HATs) in cancer [4]. Furthermore, garcinol mainly acts as an inhibitor of cellular processes like proliferation via regulation of transcription factors NF-κB and JAK/STAT3 and the PI3K/AKT signaling pathway in tumor cells and experimental animal models [5]. Besides having anti-neoplastic activity, garcinol is known to show anti-inflammatory and anti-oxidant effects [6,7]. Targeting the aforementioned signaling pathways is of great importance, as these are the pathways activated during treatment resistance. Researchers are working on developing a new garcinol nanoparticle system in order to enhance its aqueous solubility, bioavailability, and potential therapeutic effectiveness [8]. Therefore, garcinol is suitable as a therapeutic agent in combinatorial therapies where inhibition of the MAPK pathway together with the PI3K pathway is needed. The additional effect of garcinol as an inhibitor for HATs is a secondary therapeutic benefit for patients. However, use of garcinol is still in its preclinical stage, and this is mainly due to the limitations of conclusive evaluations of pharmacological parameters. Therefore, research is presently ongoing in the dimension of exploring the precise metabolic mechanism of garcinol. Numerous studies have highlighted the anti-neoplastic and pro-apoptotic potential of garcinol in different cancer entities like cervical, breast, and lung or colon carcinoma, as reviewed in [5]. The role of garcinol in regulating cadherins was first described in breast cancer. Here, treatment of cell lines with garcinol led to upregulation of the epithelial marker E-cadherin [9]. Interestingly, the upregulation of the opposing T-cadherin (CDH13) through garcinol was detected in cervical cancer [10]. Jia et al. found that garcinol dose-dependently suppressed cell viability, colony formation, invasion, migration, cell cycle progression, and promoted cell apoptosis in cell lines, as well as inhibited tumor growth in a xenograft mouse model. Importantly, the results also showed that garcinol treatment increased the expression of T-cadherin both in vitro and in vivo [11]. In summary, garcinol can create a tumor-suppressive effect through upregulation of certain cadherins in cancer. This prompted us to analyze whether garcinol could increase T-cadherin expression in melanoma and whether garcinol could therefore be used in melanoma therapy. Our working group pioneered T-cadherin research in melanoma, and we were able to show that T-cadherin is a tumor suppressor lost in melanoma followed by increased activity of the PI3K/AKT/mTOR pathway [12,13,14]. However, our previous data also show that some melanoma cell lines and also melanomas still express T-cadherin in vivo [14]. In functional assays, we showed that the re-expression of T-cadherin in melanoma cell lines decreases migration, invasion, as well as anchorage-independent growth [14]. Subcutaneous injection of melanoma cells after T-cadherin re-expression showed a reduction in the rate of tumor growth in a BALB/c nu/nu mouse tumor model [14]. T-cadherin belongs to the nonclassical cadherins, with an extracellular domain composed of five subdomains and a glycosylphosphatidylinositol moiety (GPI), which anchors it to the outer cell membrane [15]. T-cadherin has no transmembrane domain. In physiological conditions, high T-cadherin expression is detectable in the heart, vascular and nervous systems [16,17]. T-cadherin is typically not associated with cell–cell contacts; however, in migrating cells, it is located in lipid rafts or caveolae at the leading edge of the cells and stimulates Rho GTPases. Furthermore, T-cadherin is detected in the nucleus or centrosome of endothelial cells [15,16]. In this manuscript, we postulate that garcinol can inhibit melanoma cell progression through targeting different signaling pathways. We also show that the T-cadherin expression sensitizes melanoma cells for garcinol treatment. In terms of therapy, this means that T-cadherin-positive patients could be particularly receptive to garcinol therapy.

## 2. Materials and Methods

### 2.1. Cell Lines and Cell Culture Conditions

Melanoma cell lines Mel Im and Mel Juso were isolated from melanoma metastases and cultivated in Dulbecco’s modified Eagle’s medium (DMEM) supplemented with penicillin (400 units/mL), streptomycin (50 μg/mL), and 10% fetal calf serum (all from Sigma-Aldrich, Steinheim, Germany). WM3211 and SBcl2 were isolated from a primary tumor and cultivated in MCDB153 (Sigma-Aldrich, Steinheim, Germany) with 20% Leibovitz’s L-15 (PAA Laboratories, Coelbe, Germany), 2% FCS, 1.68 mM CaCl_2_ (Sigma-Aldrich, Steinheim, Germany), and 5 µg/mL insulin (Sigma-Aldrich, Steinheim, Germany). All cell lines were incubated at 37 °C in a 5% CO_2_, humified atmosphere [18]. All experiments were performed with mycoplasma-free cells (MycoSEQ mycoplasma detection system, Thermo Fisher Scientific, Waltham, MA, USA). Garcinol (Cay10566-5, Cayman Chemical, Biomol GmbH, Hamburg, Germany) treatment was performed with a stock solution of 5 mM dissolved in DMSO. Incubation time was 24 h. The final treatment concentrations are mentioned at the respective experiment or in the figure legend.

### 2.2. RNA Isolation and Reverse Transcription

Isolation of total cellular RNA from cultured cells was performed with the E.Z.N.A.^®^ Total RNA Kit I (Omega Bio-Tek, Norcross, GA, USA) according to the manufacturer’s instructions. Generation of cDNAs by reverse transcription (RT) reaction was performed with the Superscript^®^ II Reverse Transcriptase Kit (Thermo Fisher Scientific Inc., Waltham, MA, USA), which was used according to the manufacturer’s instructions.

### 2.3. Analysis of mRNA Expression

For the quantitative real-time polymerase chain reaction (q-RT-PCR), the Lightcycler^®^ II 480 SYBR Green I Master Kit (Roche Diagnostics GmbH, Mannheim, Germany) was used. The resulting PCR products are detected here with the SYBR Green I dye. The qRT-PCR was performed in 96-well microtiter plates (LightCycler^®^ 480 Multiwell Plate 96, Roche Diagnostics GmbH) for each gene of interest in duplicates. qRT-PCR analysis of gene expression was performed on a LightCycler 480 system (Roche Diagnostics GmbH). Primers (Sigma Aldrich, Steinheim, Germany) listed in Table 1 were used for gene expression analysis. Relative quantification of gene expression was performed using the ΔΔCt-method.

### 2.4. Western Blot

Cell debris was removed by centrifugation (13,000 rpm, 10 min, 4 °C), and the pellets were collected. As described previously [19], cell pellets were lysed in 100 µL Radioimmunoprecipitation Assay buffer (RIPA; Roche Diagnostics GmbH) for 15 min at 4 °C. Next, 20 to 40 µg of total RIPA lysates was loaded onto polyacrylamide gels. After separation, the gel was blotted onto a PVDF membrane. Each blot was blocked for 1 h with 5% (skimmed milk powder) SKM/TBS-T and incubated overnight at 4 °C. Primary antibodies used in this study are listed in Table 2. Primary antibodies were diluted in 3% SKM/TBS-T. After washing three times with TBS-T, the membrane was incubated with a horseradish peroxidase-coupled secondary antibody (anti-rabbit HRP or anti-mouse HRP, Cell Signalling Technology; 1 in 2000 dilution) for 1 h. The immunoreactions were visualized by ECL staining (Bio-Rad, Feldkirchen, Germany). The densitometry was performed using the LabImage program (version 4.2.3, Kapelan Bio-Imaging GbmH, Leipzig, Germany). All Western blots were without statistical evaluation, since individual measurements from individual PVDF membranes can deviate strongly from each other. 

### 2.5. SiRNA and Plasmid Transfection

A total of 2 × 10^5^ cells/well were seeded into 6-well plates and transfected with a siPool against CDH13 (functionally confirmed by siTOOLs Biotech, Planegg/Martinsried, Germany [20]) by using Lipofectamine RNAiMAX reagent (Life Technologies, Darmstadt, Germany), as described previously [21]. Si-RNA-Pools consist of multiple siRNAs resulting in efficient target gene knockdown with minimal off-target effects. Then, 50 pmol of CDH13 siRNA (siCDH13) or negative control siRNA-Pool was added to cells. Cells were incubated for 72 h for optimal efficiency.

### 2.6. Proliferation with the xCELLigence System

The xCELLigence system (Roche Diagnostics GmbH, Mannheim, Germany) is based on measurement of electrical impedance and permits real-time analysis of migration and proliferation. E-plates were used, and basic protocols recommended by the manufacturer were followed. A total of 2 × 10^3^ cells/well were counted for the proliferation measurement. Impedance is represented by a relative and dimensionless parameter named the cell index (CI). CI values = Zi-Z0/15[Ohm], where Z0 = impedance at the start of the experiment and Zi = impedance at individual time points during the experiment. The normalized cell index (NCI) was calculated as the cell index CIti at a given time point (ti) divided by the cell index CInml_time at the normalization time point (nml_time). The slope is used to describe the steepness of a curve within a given time window (in our case: ~120 h proliferation) [18].

### 2.7. Clonogenic Assay (Proliferation)

The in vitro cell survival assay based on the ability of a single cell to grow into a colony was performed, as described elsewhere [22]. In our case, 500 cells/well of a 6-well plate were plated for 17 days.

### 2.8. Statistical Analysis

Statistical analysis was performed using GraphPad Prism software 8.0 (GraphPad Software, Inc., San Diego, CA, USA). This software was also used to create the graphs. The results are calculated as the mean ± standard deviation (SD) or percentages. A *p*-value < 0.05 (*: *p* < 0.05) was considered statistically significant (n.s.: not significant; **: *p* < 0.005; ***: *p* < 0.001). The densitometry of Western blots is performed using the LabImage software (Version 4.2.3, Kapelan Bio-Imaging GbmH, Leipzig, Germany). All Western blots are shown without statistical evaluation, since individual measurements from individual PVDF membranes can deviate strongly from each other.

## 3. Results

### 3.1. Garcinol Reduces Proliferation of Different Melanoma Cells

To test for cell toxicity and the proliferative effects of garcinol, we performed clonogenic assays using the melanoma cell lines Mel Juso and Mel Im generated from melanoma metastases, as well as SBcl2 isolated from a primary tumor in the radial growth phase. Here, we could see that Mel Juso cells showed the lowest sensitivity to garcinol, as the number of colonies remained completely unaffected after treatment with 5 µM and even 10 µM garcinol, compared to Mel Im and SBcl2 cells. In Mel Im and SBcl2 cells, we observed a drastic decrease in the number of colonies after treatment with 10 µM garcinol. The size of the colonies was not significantly influenced by treatment with doses of 5 µM garcinol in any of the three cell lines. A concentration of 10 µM garcinol led to complete proliferation arrest in SBcl2 cells, whereas Mel Juso and Mel Im cells were not affected by the treatment at all (Figure 1A). Additionally, we checked cell growth in a real-time cell analysis (RTCA) assay. Similarly to the clonogenic assay, Mel Juso showed the lowest sensitivity to garcinol. Mel Im and SBcl2 cells responded with a significant decrease in proliferation at a concentration of 5 µM garcinol (Figure 1B). SBcl2 cells responded with a significant decrease in proliferation at a concentration of 5 µM of garcinol and stopped growth completely at 10 µM garcinol (Figure 1B).

### 3.2. Garcinol Induces Apoptosis in SBcl2 Cells

To confirm the functional reduction in proliferation, we performed Western blot analysis to detect an increase in expression of the cyclin-dependent kinase inhibitor p21 (WAF1/CIP1) (Figure 2A). Here, we could detect an induction in p21 protein expression after stimulation with 10 µM garcinol in all of the tested cell lines, with the highest induction in SBcl2 cells. Upon treatment with garcinol, we noticed a strong detachment of the SBcl2 cells; therefore, we followed up with measurements of the apoptosis rate by FACS analysis. Here, we could detect that Mel Juso and Mel Im cells are not prone to apoptosis induction after garcinol treatment. In line with our data, the cell line SBcl2 showed significant induction of apoptosis after being subjected to 10 µM garcinol (Figure 2B). SBcl2 cells also showed drastically increased levels of cleaved PARP (c-PARP) and cleaved CASPASE 9 (c-CASPASE9), whereas Mel Juso and Mel Im showed low levels of proteolytically cleaved proteins (Figure 2C), confirming the data obtained through FACS analysis.

### 3.3. Expression of T-Cadherin (CDH13) Is Not Influenced by Garcinol Treatment

We continued with an investigation of the particularly strong effect of garcinol on the SBcl2 cell line. Since there was literature suggesting a distinct role of T-cadherin (CDH13) during garcinol treatment [10], we analyzed the T-cadherin protein status in additional melanoma cell lines next to the melanoma cell lines used here in previous functional assays. Interestingly, the most garcinol-sensitive SBcl2 showed a higher T-cadherin protein level than the other more garcinol-insensitive melanoma cell lines used. The cell line D04, transfected with a T-cadherin over-expressing vector [14], was used as a positive control. WM3211 also showed T-cadherin expression, whereas all other melanoma cell lines harbor no T-cadherin (Figure 3A). Next, we tested whether garcinol also leads to an increase in T-cadherin expression. A slight induction of this cell–cell adhesion molecule was detected at mRNA and protein level (Figure 3B,C). For the next experiments, we established T-cadherin siRNA experiments and found that the efficiency of the knockdown was best after 72 h (Appendix A). We further combined knockdown experiments and garcinol treatment in SBcl2 and detected continued CDH13 knockdown when simultaneously treated with garcinol at concentrations of 5 to 10 µM. To further investigate the effect of garcinol on SBcl2 cells, we checked for CDH-2 (N-Cadherin) expression after knockdown of CDH13. Interestingly, we detected an increase in CDH-2-expression at highest garcinol concentrations (Figure 3D). In summary, we were able to show that garcinol treatment leads to a slight increase in CDH13 expression, yet cannot overcome siRNA knockdown of CDH13. However, CDH-2 expression is induced after garcinol treatment and simultaneous CDH13 knockdown, hinting towards an ongoing cancerous cellular phenotype, as CDH-2 is a marker for melanoma progression.

### 3.4. The Role of T-Cadherin for Garcinol Treatment

To determine whether T-cadherin expression has an impact on the efficacy of garcinol, we performed siRNA-mediated knockdown in T-cadherin-positive SBcl2 cells and investigated proliferation and apoptosis in these cells. With an increase in the garcinol concentration (10 µM) and constant CDH13 knockdown (siCDH13), we detected a reduction in the amount of p21 protein expression in the cells (Figure 4A). In line with these data, the protein level of Cyclin D1 was not drastically influenced by garcinol when T-cadherin is reduced by siRNA. In any case, Cyclin D1 was not reduced by garcinol when T-cadherin knockdown is present, proving that cells with less T-cadherin simply continue to proliferate despite treatment with garcinol (Figure 4B). Next, we performed RTCA assays and were able to confirm the continued proliferation of SBcl2 cells when garcinol is administered at a dose of 5 µM and 10 µM and T-cadherin expression is reduced (white bars). In comparison, T-cadherin-expressing (ctrl) cells no longer proliferate, but apparently seem to die (black bars) (Figure 4C). To analyze the influence of garcinol and siCDH13 on apoptosis, we performed Western blot analysis and revealed that the expression of c-CASPASE9 and c-PARP are both reduced when the siCDH13-treated cells are subjected to 10 µM garcinol (Figure 4D,E). The tumor suppressor protein p53, encoded by the TP53 gene, is a transcription factor that activates DNA repair mechanisms and apoptosis. We revealed that the protein level of p53 was also reduced by siCDH13 transfection and treatment with garcinol (Figure 4F). We therefore concluded that despite an increased dose of garcinol, less apoptosis occurs because the downregulation of T-cadherin by siCDH13 transfection is de-sensitizing cells to garcinol-mediated effects. To confirm this hypothesis, we performed FACS analyses, where we could reveal that the reduced expression of T-cadherin reduced the number of apoptotic cells, even with higher doses of garcinol (Figure 4G).

### 3.5. A Loss of T-Cadherin Influences the Effect of Garcinol Treatment

We further investigated the role of T-cadherin in garcinol treatment in the endogenously T-cadherin-positive cell line WM3211, even if the mode of action of garcinol together with downregulation of T-cadherin is less pronounced. In WM3211 we performed a T-cadherin (siCDH13) knockdown and treated the cells with 5 µM and 10 µM garcinol. With a concentration of 10 µM garcinol and siCDH13 transfection, the cells showed less p21 protein expression compared to the transfected control cells (Figure 5A). Quantification of the protein expression can be found in Appendix A. In principle, the downregulation of T-cadherin causes an increased expression of Cyclin D1, regardless of whether high concentrations of garcinol are used. A combination of garcinol treatment and knockdown of CDH13 led to a less pronounced induction of the Cyclin D1 protein quantity (Figure 5B). Especially when the cells were incubated with a concentration of 5 µM garcinol after siCDH13 transfection, the proliferation rate in the RTCA increased significantly (Figure 5C). Regarding the prevention of apoptosis, we could not find any regulation of c-CASPASE9, c-PARP, p53, and Bcl-xL with low T-cadherin protein level and garcinol treatment combined (Figure 5D–H). Quantification of the Western blots can be found in Appendix A. This is probably related to a Cysteine (C)-to-Glycine (G) mutation of the amino acid 242 of the tumor suppressor TP53 in WM3211 cells, which are generally not very sensitive to apoptosis. In summary, the analysis of SBcl2 and WM3211 shows that reduced expression of CDH13 de-sensitized the cells to the treatment with garcinol.

## 4. Discussion

In this study, we investigated the complex role of T-cadherin (CDH13) with a focus on regulating the sensitivity to garcinol in human melanoma cells. Garcinol, isolated from the plant also known as mangosteen, is a viable option for cancer treatment. The polyisoprenylated benzophenone garcinol has already been shown to have an inhibitory effect on hyper-activated signaling pathways in cancer [23,24]. If surgery for malignant melanoma is not possible or not sufficient, or if the tumor has already metastasized, targeted therapy and immunotherapy are currently available. Targeted therapies with chemotherapeutic agents and immunotherapy with checkpoint inhibitors are among the systemic therapies recommended in the guideline for clinicians [25]. Although there have been constant improvements in the treatment of melanoma, resistance to these therapies continues to occur [26]. It is therefore urgently necessary to give new therapies a chance and to conduct research into new reagents. Garcinol mainly acts as an inhibitor of cellular processes like proliferation via regulation of transcription factors, such as NF-κB. It has been shown to interact with the JAK/STAT3 and PI3K/AKT signaling pathways as well in tumor cells, which has already been proven in experimental animal models [5]. Aggarwal and colleagues summarized the potency of garcinol to inhibit histone acetyltransferases (HATs) in cancer [4,5]. Therefore, garcinol is suitable as a therapeutic agent in combinatorial therapies, in which inhibition of the MAPK pathway together with the PI3K pathway is relevant. The HAT-inhibitory effect of garcinol might be a second therapeutic benefit for patients.

However, clinical use of garcinol is still in its pre-clinical stage, and this is mainly ascribed to the limitations of conclusive evaluations of pharmacological parameters. Therefore, the purpose of this study was to explore the function of garcinol with regards to specific cancer properties such as proliferation and apoptosis in melanoma cells in more detail. In this present study, we showed that melanoma cells, treated with different concentrations of garcinol (5 and 10 µM), dose-dependently suppressed cell viability and colony formation and induced apoptosis. We were therefore interested in the mechanisms of garcinol action in melanoma cells and focused on the functional properties of proliferation and apoptosis. Our data show that the 10 µM concentration of garcinol has an inhibitory effect on cell growth and induces apoptosis in some of the tested melanoma cells in vitro. Quite fittingly, Ye et al. have demonstrated that garcinol suppresses the proliferation of the breast cancer cell line MCF-7 promoted by 17β-estradiol [27]. It was further shown that garcinol inhibits cell proliferation, angiogenesis, cell cycle progression, and induces apoptosis in oral squamous cell carcinoma (OSCC) [5,28], as well as in human leukemia HL-60 cells [29]. Interestingly, in our present study, we observed strong differences in sensitivity to garcinol in our tested cell lines. The metastatic melanoma cell line Mel Juso required higher doses of garcinol to show an inhibitory effect on proliferation, compared to the very sensitive primary tumor cell line SBcl2. In subsequent analyses, we checked for differently regulated genes in these cell lines and revealed that the expression status of CDH13 was directly correlated with the sensitivity to garcinol in melanoma cells, which has also been described by Zhao et al. (2018) for human cervical cancer cells [10]. The more garcinol-sensitive SBcl2 cell line expresses CDH13, whereas the more insensitive Mel Juso and Mel Im cells do not express CDH13. This aspect is highly interesting, since we could already show in previous studies about T-cadherin expression in melanoma that 80% of analyzed melanoma cell lines lose T-cadherin expression compared to normal melanocytes (NHEM), and 70% of analyzed and microdissected human melanoma tissues derived from melanoma patients showed no T-cadherin expression either [14]. This opens up new perspectives for garcinol’s usage in clinical treatment. In different cancer entities (e.g., lung, ovarian, esophageal, bladder, cervical, prostate), CDH13 was already suggested to be a tumor-suppressive factor, and its downregulation was associated with tumor growth and metastasis [15]. Furthermore, garcinol treatment led to induced expression of CDH13 in Mel Im and SBcl2 cells, but not in Mel Juso cells. There is also further evidence from previous studies that garcinol can have an influence on markers of the epithelial–mesenchymal transition, which also include other adhesion molecules, such as E-cadherin (CDH-1) and vimentin [30,31]. For this reason, we also analyzed the expression of CDH13 and N-cadherin (CDH-2) after knockdown of CDH13 and subsequent treatment of these cells with garcinol. CDH13 was not re-induced by garcinol in the CDH13-deficient cells; however, CDH-2 was induced at a garcinol concentration of 10 µM. It should be noted that CDH-2 is considered a tumor inducer in melanoma, hinting towards a more aggressive phenotype of the cells. Therefore, CDH-2 is also a marker for the loss of the garcinol effect after CDH13 reduction [32,33]. After performing an siRNA-mediated knockdown of CDH13 in the cell lines SBcl2 and WM3211 (both endogenously expressing CDH13), we achieved de-sensitization of the cell lines to garcinol treatment. Due to CDH13 knockdown, high concentrations of garcinol lead to ongoing cell proliferation, as measured in RTCA experiments and confirmed by the downregulation of p21 and continued expression of Cyclin D1. Furthermore, apoptosis was reduced under these conditions, as proven by FACS analyses. Additionally, we confirmed the prevention of apoptosis after CDH13 knockdown and under garcinol treatment, since less cleaved CASPASE9 and PARP were detected, along with TP53 being reduced. It should be noted that the WM3211 cell line did not show a drastic or even any de-sensitization in apoptosis assays because the cell line harbors a TP53 mutation (C242G) (Cellosaurus WM3211 (CVCL_6797)). We therefore postulate that cells with a low CDH13 expression which are treated with garcinol seem to undergo a switch in phenotype to a more aggressive cellular state, proven by the re-expression of CDH-2. This aspect confirms that a garcinol pre-treatment screening for CDH13 expression might be useful. Molecular diagnostics are increasingly performed routinely in the diagnosis and management of patients with melanoma [34]. We believe that CDH13 expression might be a novel biomarker for the possibility of a garcinol treatment. Further research is needed to uncover the precise mechanisms and signaling pathways involved in garcinol and CDH13 pharmacodynamics and the implication for CDH13 screening in melanoma patients. In summary, garcinol treatment was shown to inhibit proliferation and induce apoptosis in melanoma. In addition, CDH13 expression in melanoma cells sensitizes to garcinol treatment. If melanoma cells do express CDH13, they are less sensitive to garcinol in proliferation and FACS analyses. In melanoma cell lines expressing a small amount of CDH13, garcinol treatment can lead to an increase in the expression of CDH-2, indicating a pro-tumorigenic phenotype of the cells. This means that, in vivo, patients with CDH13 expression in particular could benefit from garcinol treatment. And if resistance occurs with other therapies, garcinol therapy could be successful, especially in CDH13-positive patients. However, further in vivo studies are needed to strengthen this hypothesis.

## 5. Conclusions

In conclusion, garcinol treatment was shown to inhibit proliferation and induce apoptosis in some melanoma cell lines. In addition, CDH13 expression in melanoma sensitizes to garcinol treatment. If melanoma cells do not express CDH13, they are less sensitive to garcinol in proliferation and FACS analyses. If melanoma cell lines express little CDH13, garcinol treatment can lead to an increase in the expression of CDH2, indicating a pro-tumorigenic phenotype of the cells. This means that, in vivo, patients with CDH13 expression in particular could be susceptible to garcinol. Or if resistance occurs with other therapies, garcinol therapy could be successful, especially in CDH13-positive patients.

## Figures and Tables

**Figure 1 cancers-16-01853-f001:**
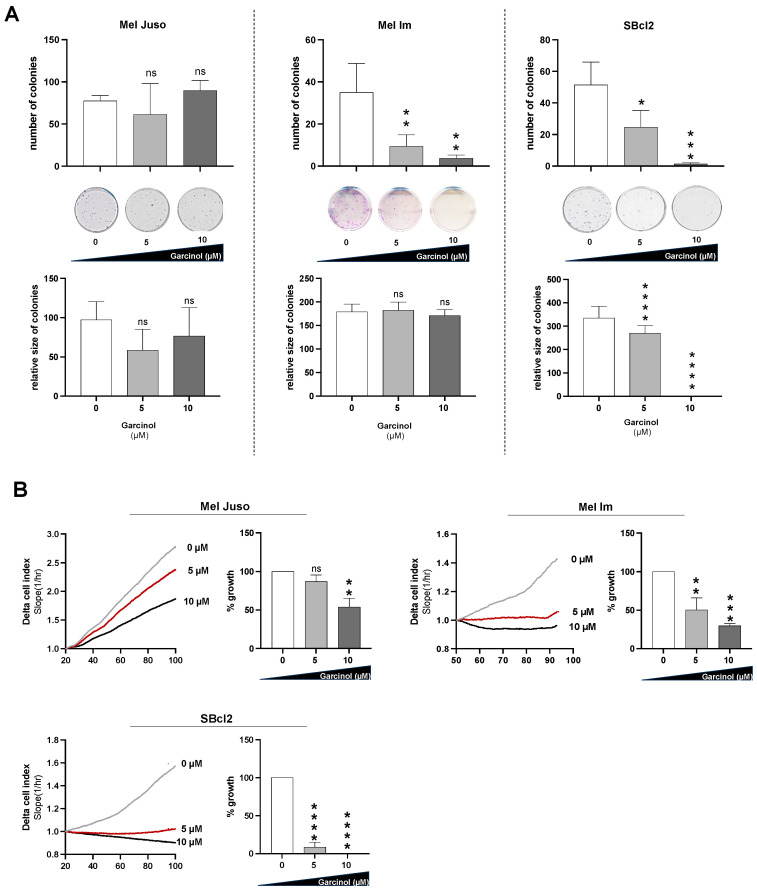
(**A**) Clonogenic assay analyzing the clonogenic ability defined by the number and size of colonies. Mel Juso, Mel Im, and SBcl2 melanoma cells were treated with 5 µM and 10 µM garcinol, respectively. As a control (0 µM), cells were treated with DMSO, the diluent of garcinol. Representative images of colonies 17 days after seeding, fixation, and staining using crystal violet (*n =* 3). (**B**) RTCA analysis detecting proliferation. The total proliferation time which was analyzed is 100 h. The quantification of the proliferation is normalized to the diluent control (*n =* 3) (Delta Cell index  *=*  relative change in measured impedance to represent cell status). Statistical analysis was performed using one-way ANOVA followed by Tukey’s multiple comparison tests (ns: not significant; *: *p* < 0.05; **: *p* < 0.005; ***: *p* < 0.001; ****: *p* < 0.0001). All bar graphs are shown as mean ± standard deviation (SD).

**Figure 2 cancers-16-01853-f002:**
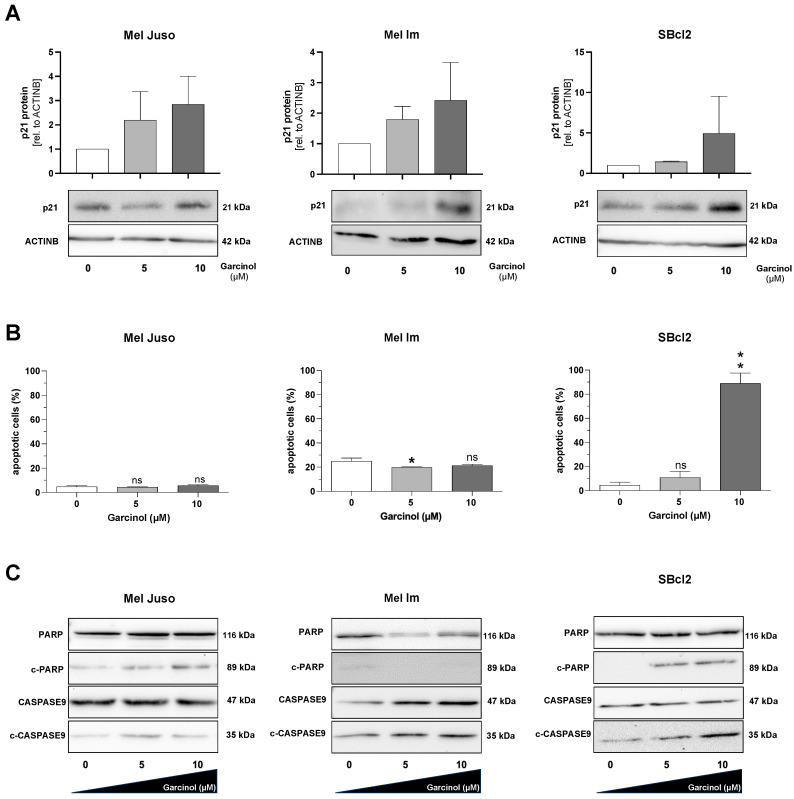
(**A**) Representative Western blot images depicting protein expression level of p21 (CIP1/WAF1) (21 kDA). Equal loading was controlled by ACTINB (42 kDA) primary antibody. Control (0) set as 1. (**B**) FACS analysis for measuring apoptotic cells. The quantification of the early and late apoptotic cells was integrated in one graph (*n =* 3). (**C**) Representative Western blot images depicting the analysis of apoptosis via the protein levels of PARP, cleaved-PARP (c-PARP), CASPASE9 and cleaved-CASPASE9 (c-CASPASE9). Quantification of cleaved proteins in relation to the whole protein amount of PARP or CASPASE9. Experiments were performed at least three independent times. Statistical analysis was performed using one-way ANOVA followed by Tukey’s multiple comparison tests (ns: not significant; *: *p* < 0.05; **: *p* < 0.005). All bar graphs are shown as mean ± standard deviation (SD). The uncropped blots are shown in Appendix A.

**Figure 3 cancers-16-01853-f003:**
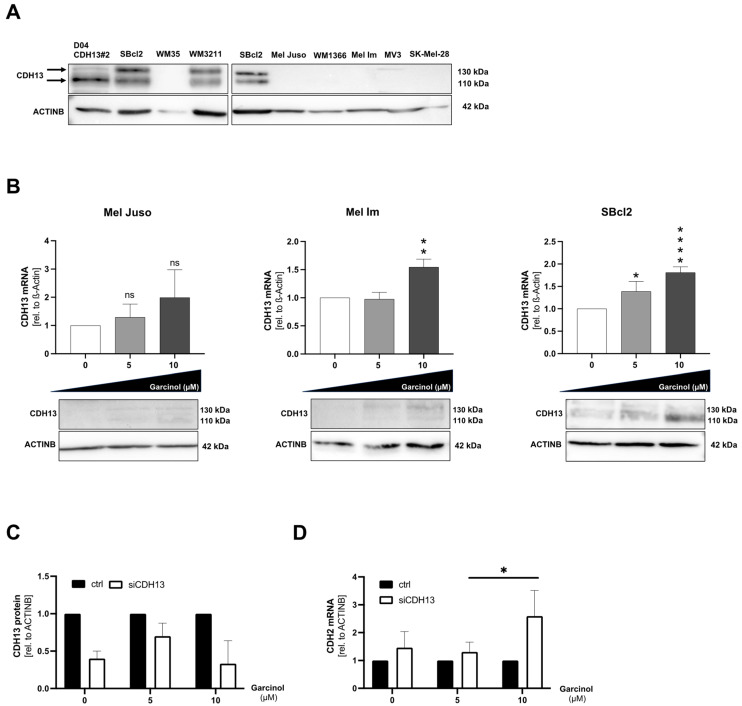
(**A**) Representative Western blot images depicting protein expression of CDH13 in eight different melanoma cell lines (SBCl2, WM35, WM3211, Mel Juso, WM1366, Mel Im, MV-3, SK-Mel-28). First lane shows the CDH13-positive control, which was stably transfected with a CDH13 expression construct (D04 CDH13#2), as described previously [14]. SBcl2 was shown twice as internal Western blot control because two different blots are shown. ACTINB was used as the loading control. (**B**) Analysis of mRNA levels of CDH13 relative to β-actin in qRT-PCR (*n =* 3). (**C**) Protein expression of CDH13 after garcinol treatment. ACTINB served as the loading control. The graphs show the densitometry of three different experiments. ACTINB was used as the loading control. (**D**) Analysis of mRNA levels of CDH-2 relative to β-actin in qRT-PCR (*n =* 3). Statistical analysis was performed using one-way ANOVA followed by Tukey’s multiple comparison tests. Statistical analysis of Figure 3D, which includes multiple variables, was performed using two-way ANOVA followed by Sidak’s multiple comparison tests (ns: not significant; *: *p* < 0.05; **: *p* < 0.005; ****: *p* < 0.0001). All bar graphs are shown as mean ± standard deviation (SD). The uncropped blots are shown in Appendix A.

**Figure 4 cancers-16-01853-f004:**
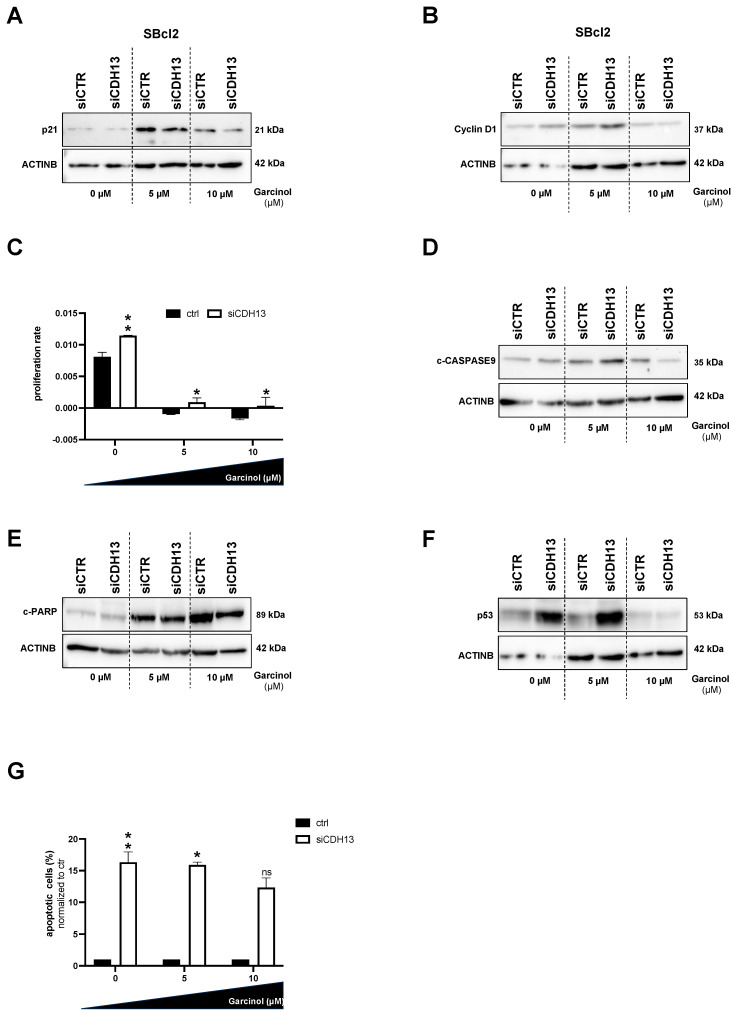
(**A**,**B**,**D**–**F**) Representative Western blot images depicting protein expression of p21 (**A**), Cyclin D1 (**B**), c-CASPASE9 (**D**), c-PARP (**E**), and p53 (**F**) in SBcl2 cells transfected with siCDH13 and siCTR, respectively, and subsequently treated with 0, 5, and 10 µM garcinol for 24 h. Densitometry of CDH13 protein expression in relation to ACTINB as loading control (Appendix A). Black bars represent the siCTR transfected, white bars represent the siCDH13 transfected cells (*n* = 3). (**C**) Real-time cell proliferation analyses using the xCELLigence system (RTCA) (*n* = 3). (**G**) Quantification of flow cytometric detection of apoptotic cells (*n* = 3). Statistical analysis of Figure 4C,G, which include multiple variables, was performed using two-way ANOVA followed by Sidak’s multiple comparison tests (ns: not significant; *: *p* < 0.05; **: *p* < 0.005). All bar graphs are shown as mean ± standard deviation (SD). The uncropped blots are shown in Appendix A.

**Figure 5 cancers-16-01853-f005:**
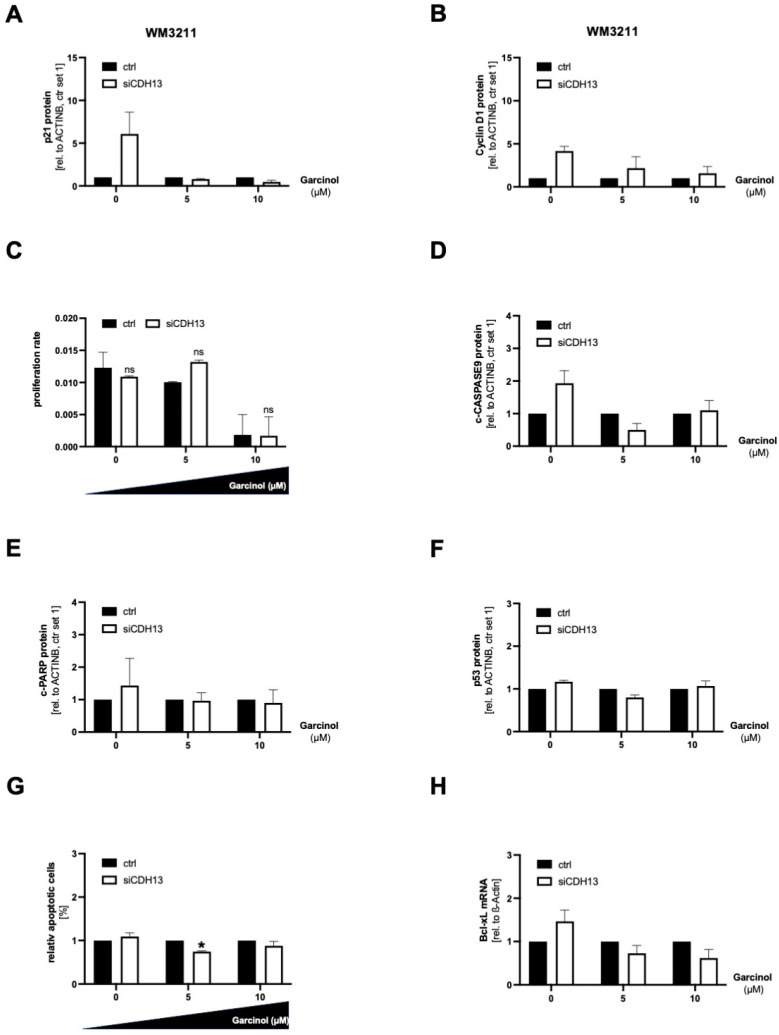
For all experiments, WM3211 melanoma cells were transfected with siCDH13 for 72 h and subsequently treated with 0, 5, or 10 µM garcinol for 24 h. (**A**) Representative Western blot images and corresponding densitometry of p21 expression. ACTINB served as a loading control. (**B**) Representative Western blot images and corresponding densitometry of Cyclin D1 expression. ACTINB served as a loading control. (**C**) Real-time cell proliferation analyses using the xCELLigence system (RTCA). (**D**) Representative Western blot images and corresponding densitometry of cleaved CASPASE9 (c-CASPASE9) expression. ACTINB served as a loading control. (**E**) Representative Western blot images and corresponding densitometry of cleaved PARP (c-PARP) expression. ACTINB served as a loading control. (**F**) Representative Western blot images and corresponding densitometry of p53 expression. ACTINB served as a loading control. (**G**) Quantification of flow cytometric detection of apoptotic cells (*n* = 3). (**H**) Representative Western blot images and corresponding densitometry of Bcl-xL expression. ACTINB served as a loading control. (ns: not significant; *: *p* < 0.05). All bar graphs are shown as mean ± standard deviation (SD).

**Table 1 cancers-16-01853-t001:** Oligonucleotides used as primers for real-time PCR.

Gene	Forward Primer	Reverse Primer
hβ-Actin	CTACGTCGCCCTGGACTTCGAGC	GATGGAGCCGCCGATCCACACGG
hCDH13	GGCAATTGACAGTGGCAACC	TGCAGGAGCACACTTGTACC
hCDH-2	TGGATGAAGATGGCATGG	AGGTGGCCACTGTGCTTAC
hBcl-xL	TGACCACCTAGAGCCTTGGA	TGAACAGGATACTTTTGTGGAACT

**Table 2 cancers-16-01853-t002:** Antibodies used for Western blot analysis.

Antibody	Company	Reference	Dilution
ACTINB	Sigma Aldrich, Steinheim, Germany	A5541	1:5000
CASPASE9	Cell Signalling Technology, Frankfurt a. M. Germany	#9502	1:1000
CDH13	Abcam, Berlin, Germany	ab36905	1:1000
Cyclin D1	Santa Cruz Biotechnology, Heidelberg, Germany	sc-8396	1:500
p53	Santa Cruz Biotechnology, Heidelberg, Germany	sc-126	1:2000
p21	Abcam, Berlin, Germany	ab109199	1:1000
PARP	Cell Signalling Technology, Frankfurt a. M. Germany	#9542	1:1000

## Data Availability

The data presented in this study are available in this article (and Appendix A).

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
