# Peer review of "The Role of T-Cadherin (CDH13) in Treatment Options with Garcinol in Melanoma"

_cancers, 2024, doi:10.3390/cancers16101853_

Round 1
Reviewer 1 Report
Comments and Suggestions for Authors
The authors present an interesting article about the in vitro action of garcinol on some melanoma cell lines, on which I have the following observations to make:
1. In paragraph 3.5 (line 292) results related to WM3211 cell line are presented, but this cell line is not presented in the material and methods section
2. In paragraph 4 (line 349) reference is made to treatments with garcinol in concentrations of 5, 10, and 25 µM, but in the presented materials there are no results related to the treatment with a concentration of 25 µM
3. The paragraph between lines 352-354 (Our data show that garcinol at a concentration of 10µM has an inhibitory effect on cell growth and induces apoptosis in melanoma cells in vitro) is not supported by the results presented (see Mel Juso cell line)
4. Likewise, the first sentence of paragraph 5 (In conclusion, garcinol treatment was shown to inhibit proliferation and induce apoptosis in melanoma) must be adapted to the presented results.
Reviewer 2 Report
Comments and Suggestions for Authors
1) Please mark the panels in all the figures in the whole blot pictures.
2) In fig 2A and 4A western blot, why is p21 not visible in the full blot picture? The authors must show the p21 band in the full blot with a higher exposure/increased protein loading. On the other hand, the cropped blot pictures showing p21 bands does not have the beta-actin loading control. Why is that? If the protein is expressed, then either loading more protein and/or a higher exposure would produce the bands at the correct position. Do not cut the membrane horizontally so that the entire range of markers are not visible. This defeats the entire purpose of providing whole blot pictures. Please show p21 expression in a blot that shows bands for both p21 and the loading control. Same for Fig 4, p53 and C-PARP bands. Do not cut the membrane, in these blots, loading control is not visible! Show p53/C-PARP bands in a full blot which also shows the loading control.
Reviewer 3 Report
Comments and Suggestions for Authors
The manuscript by Staebler et al. offers a comprehensive overview of the effects of garcinol on melanoma cell lines, highlighting its ability to reduce proliferation and induce apoptosis. They also found that cells that express the cell-cell adhesion protein T-cadherin (CDH13) are more sensitive to the drug garcinol.
The data presented in the manuscript is interesting. I have a few suggestions for polishing the manuscript.
Major Comments:
- The manuscript should be checked by a native English speaker.
- All the bar graphs should be expressed as mean ± SD instead of mean ± SEM.
- The authors should quantify all the western-blot images. They have quantified some western blot images. It should be uniform throughout.
- The authors showed that there is an increased expression of CDH13 in SBcl2 compared to others. Is it due to increased translation or protein stability or both?
- Did the authors verify the interaction between CDH13 and garcinol using filter binding assays?
- The authors concluded that in-vivo patients with CDH13 expression are particularly susceptible to garcinol. This is an overestimation. To make sure the data is correct, the authors should use xenograft/PDX mice with WT and siRNA CDH13 KO in the presence and absence of garcinol to see how the tumors progress. This will give some hints about the proposed mode of action in in vivo patient samples.
Minor Comments:
- In the Materials and Methods Section, the authors should explain what method (e.g., 2^-∆∆ct) they have used to calculate the level of mRNA expression.
- Replace "fragments" with "debris" on Page 3, Line 117.
- The authors should include a graphical abstract for a broader audience.
The manuscript should be checked by a native English speaker.
Round 2
Reviewer 2 Report
Comments and Suggestions for Authors
The authors have satisfactorily addressed my concerns. I would recommend accepting the revised manuscript for publication.
Reviewer 3 Report
Comments and Suggestions for Authors
The manuscript by Staebler et al. offers a comprehensive overview of the effects of garcinol on melanoma cell lines, highlighting its ability to reduce proliferation and induce apoptosis. They also found that cells that express the cell-cell adhesion protein T-cadherin (CDH13) are more sensitive to the drug garcinol.
The authors have addressed all the previous comments. Thus, the manuscript can be accepted in its present form.
Round 3
Reviewer 3 Report
Comments and Suggestions for Authors
The manuscript by Staebler et al. offers a comprehensive overview of the effects of garcinol on melanoma cell lines, highlighting its ability to reduce proliferation and induce apoptosis. They also found that cells that express the cell-cell adhesion protein T-cadherin (CDH13) are more sensitive to the drug garcinol.
The authors have addressed all the previous comments. Thus, the manuscript can be accepted in its present form.